# White Paper Assistance: A Step Forward Beyond the Shortcut Learning

## Abstract

The promising performances of CNNs often overshadow the need to examine whether they are doing in the way we are actually interested. We show through experiments that even over-parameterized models would still solve a dataset by recklessly leveraging spurious correlations, or so-called "shortcuts". To combat with this unintended propensity, we borrow the idea of printer test page and propose a novel approach called White Paper Assistance. Our proposed method is twofold; (a) we intentionally involves the white paper to detect the extent to which the model has preference for certain characterized patterns and (b) we debias the model by enforcing it to make a random guess on the white paper. We show the consistent accuracy improvements that are manifest in various architectures, datasets and combinations with other techniques. Experiments have also demonstrated the versatility of our approach on imbalanced classification and robustness to corruptions.

## 1 Introduction

*We don't see things as they are; we see them as we are.*

–An Old Proverb

These words give us insight into the predictable irrationalities of the human mind. Individuals always create their own "subjective reality" from their perception. Psychological researches (Haselton et al., 2015; Zhang et al., 2007; Shafer et al., 1984; Kahneman & Tversky, 1996) term this systematic, irrational, unconscious error that can dramatically alter the way we perceive the world as "cognitive biases". Similarly to the behavior of human, convolutional neural networks may also develop their own biases during training, by learning "shortcuts" (Geirhos et al., 2020)(also known as "spurious cues" (Hendrycks et al., 2021) or "superficial correlations" (Jo & Bengio, 2017; Pezeshki et al., 2020)) which perform well on the existing test data but would fail dramatically under more general settings.

There is a large volume of published studies describing and analysis this learning dynamic. In this work, we adopt the *gradient starvation* hypothesis, proposed in (des Combes et al., 2018; Pezeshki et al., 2020), that the leading cause for this *feature imbalance* is that the neural network is biased towards capturing statistically dominant features in the data so that it starves the learning of other very informative but less frequent features. With this being considered, a natural question is *how to favor generalizable features over shortcuts*? It seems that the most reasonable and direct way is to identify which features contain shortcuts (like green to frogs) and which features should be enhanced (like shapes to animals)? Unfortunately, most patterns that CNNs rely on to classify do not appear in a form amenable to discover. And enhancing specific features requires specific expert knowledge, let alone extensive manpower and resources.

Luckily, CNNs are not alone with this issue. Very much like the networks submitting to spurious preference, the printer sometimes may use an unintended color to represent the intended color. In the real world, we call it color cast problem. When a colored image is fed into a printer, the printer has to perceive it and then duplicate it using the right color. The color cast problem thereby indicates the wrong propensity of color using. In practical use, when we suspect that our printers are having color cast problems, we usually let the printer print a white paper. Once this white paper is printed into other colors, the color cast is thereby detected and we need to seek corresponding solutions.

Put another way, the white paper here serves as a prefect indicator of the color cast problem. This common sense motivates us to exploit the use of white paper to regularize the model.

Intuitively, the white paper does not belong to any classes the model have learned from whichever benchmark dataset. A idealized model should thereby give an inference result that is almost as if it makes a random guess, to demonstrate it does not mistake this sample as any class it has learned. Consequently, when discovering a difference between the intended and actual outcome, we could know that the model now has some unintended generalization directions, which should be thought of as a consequence of shortcut learning. Simply put, the white paper could also act like a "test paper" in detecting dominant patterns. The experimental results in Section 3 will prove that the use of the white paper is an effective and universal choice, even when compared with some real datasets.

Leveraging the superior ability of the white paper in detecting dominant patterns, we derive an interesting and effective regularizer called White Paper Assistance, which alleviates the excessive reliance of these dominant features by repeatedly enforcing the model to make a random guess on the white paper. Our method does not require any further supervision on the bias, such as explicit labels of misleadingly correlated attributes (Kim et al., 2019; Li & Vasconcelos, 2019; Sagawa et al., 2019), or domain-specific-bias-tailored training technique (Wang et al., 2019; Geirhos et al., 2019; Li et al., 2020). Moreover, despite the simplicity of implementation of the white paper, our method can effectively improve the model's generalization ability and help produce better performance. Since the whole algorithm does not entail any modification on model architectures and any interference to original training, it can be easily assembled into various CNNs as "plug-and-play" components, which significantly promotes its value in practice.

Here we summarize our contributions:

- We propose a novel method called White Paper Assistance to alleviate the shortcut learning. Our method does not require modifying the network and is easily implementable on any modern neural architecture.

- We show the superior ability of the white paper in detecting dominant patterns.

- We experiment with various architectures, different benchmark datasets, different combinations of techniques to show the wide applicability and compatibility of our method.

- We test our method in imbalanced classifications and robustness against corruptions to demonstrate the versatility.

## 2 RELATED WORK

With the emergence of deep learning, numerous astonishing stories (He et al., 2016; Chen et al., 2017; Yun et al., 2019; Radosavovic et al., 2020) about tremendous performances of CNNs have rapidly spread all over the field. However, despite the ever-increasing pace, CNNs share the same vulnerability as the human cognitive system, bias. It has been demonstrated that models may learn spurious shortcut correlations, which may be sufficient to solve a training task but are clearly lack of generalization utility. For example, a model would identify cows in "common" (*e.g.* pastures) contexts correctly but fail to classify cows in "uncommon" (*e.g.* beach) contexts (Beery et al.). Standard ImageNet-trained models prefer to label a cat image with elephant skin texture as elephant instead of cat (Geirhos et al., 2019). Such phenomenons (Nguyen et al., 2014; Wichmann et al., 2010; Ribeiro et al., 2016) highly exemplify the contradiction between the shortcut correlations and the human-intended generalization.

As an active line of research, numerous studies have provided different explanations for this phenomenon (Nasim et al., 2019; Xu et al., 2019; Parascandolo et al., 2020). For example, Valle-Perez et al. (2019) suggests that the parameter-function map of networks would bias towards simple functions. Kalimeris et al. justifies the simplicity bias further by showing that SGD learns functions of increasing complexity. Hermann & Lampinen (2020) demonstrates the model being "lazy" that it would favor the easier-to-extract feature over a more predictive feature. In this paper, we follow the explanation proposed in (des Combes et al., 2018; Pezeshki et al., 2020) and argue that the rationale behind this learning proclivity for shortcuts is the propensity of the model to capture statistically dominant features in the data, rendering failure on discovering other predictive features.

Recently studies that relates to shortcut removal usually requires extra supervision (Kim et al., 2019; Sagawa et al., 2019). Li & Vasconcelos (2019) explicitly add color bias as side information to an unbiased dataset of grayscale images. Geirhos et al. (2019) use style transfer to synthesize data to help generate a more preferable shape-based representation. Li et al. (2020) further provide supervisions from both shape and texture when generating cue conflict images and lead to better feature representations. Instead of leveraging laborious and expensive supervision, our method utilizes the common sense by leveraging the white paper to detect the dominant patterns.

# 3 OUR METHOD

The algorithm we propose is a conceptually simple and plug-and-play method that can be easily integrated into various CNN models without changing the learning strategy. The pseudo-code of the White Paper Assistance is shown in Algorithm 1. Generally speaking, the aim of this algorithm is to *detect and conquer*.

**Detect:** For certain epoch from training iteration, the probability to conduct White Paper Assistance is $P$, and $1 - P$ if otherwise. Once applying it, a batch of white paper will be fed into the model and we can obtain the normalized output distribution (using "softmax") $\boldsymbol{p}$. The distribution here represents the perception of the model for this white paper and, more importantly, the model's propensity for unintended patterns.

**Conquer:** As we've discussed before, since the white paper does not belong to any class the model has learned, it should give an inference result that is almost as if it makes a random guess, to demonstrate it is not biased towards any pattern. In the case of the multi-class classification with $N$ classes, the ideal prediction probability distribution for the white paper would be $\boldsymbol{q} = [\frac{1}{N}, \frac{1}{N}, \frac{1}{N}, ..., \frac{1}{N}]$. Hence to measure the match of these two predictions $\boldsymbol{p}$ and $\boldsymbol{q}$, we adopt the Kullback-Leibler Divergence:

$$\mathcal{L}_{wp} = \lambda * D_{KL}(\boldsymbol{p}\|\boldsymbol{q}) \tag{1}$$

where $\lambda$ denotes the strength of the White Paper Assistance. Then we repeat this process for $M$ iterations in the hope of alleviating this unintended propensity. [1]

---

**Algorithm 1** Pseudo-code of the White Paper Assistance

---
1: **for** each epoch **do**
2:     *Real Images* training using original loss function
3:     Update model parameters
4:     initialize $p \leftarrow Rand(0, 1)$                    ▷ White Paper Assistance starts here.
5:     **if** $p < P$ **then**
6:         **for** each *iteration* $\in [1, M]$ **do**
7:             Generate a batch of white picture $W$
8:             $\boldsymbol{p} \leftarrow \text{Model}(W)$                         ▷ *White paper* training.
9:             Update model parameters by Eq. (1)
10:        **end for**
11:    **end if**                          ▷ White Paper Assistance ends here.
12: **end for**

---

There are two important questions for designing above Algorithm:

> Q1.    Does the White Paper Assistance indeed alleviate the shortcut learning?

> Q2.    Why choose using the white paper?

To answer the *first question*, we evaluated our method in a controlled experimental setup, by adding synthetic shortcuts to the data. Specifically, we added a 4×4 black square block on the top left corner of each training and testing sample of the first class (apple) of CIFAR100 (We refer to this modified dataset as **Shortcut-CIFAR100**). When trained on Shortcut-CIFAR100, this small block allows a network to achieve a negligible loss by only learning to discriminate this block on the same position while ignoring other information. Therefore, after training on Shortcut-CIFAR100,

---
[1] We explain the reason for repeating in Appendix. B.

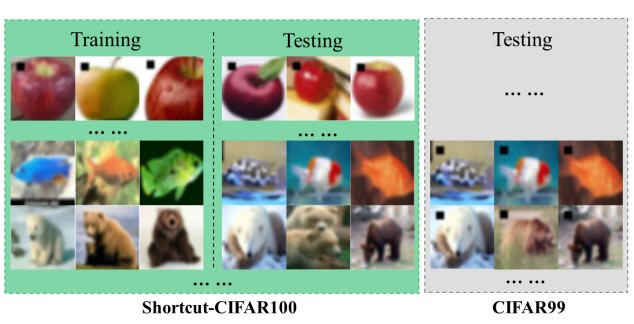 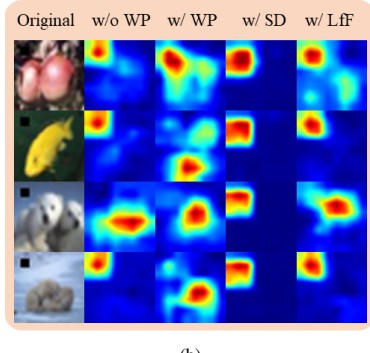

(a)                                          (b)

Figure 1: **Conceptual experiments: (a)** Illustration of Shortcut-CIFAR100 and CIFAR99. On Shortcut-CIFAR100, all the samples of "apples" are modified, including training and testing samples. On CIFAR99, all the testing samples except for samples of "apples" are modified. **(b)** Class activation mapping (CAM) visualizations of models trained on Shortcut-CIFAR100 on samples with synthetic shortcuts. While other models fixate on added blocks, WP(be short for White Paper Assistance) successfully alleviates the excessive reliance of the model on shortcuts and help it capture more informative features. We provide more visualizations in Appendix. A.1.

Table 1: **Proof of concept**: Top-1 accuracy(%) on Shortcut-CIFAR100 and CIFAR99. All the models(ResNet-56) were trained on Shortcut-CIFAR100.

| Model | Shortcut-CIFAR100 | CIFAR99 |
|---|---|---|
| ResNet-56 | 62.46 | 39.95 |
| ResNet-56 w/ WP | **66.80** | **46.88** |
| ResNet-56 w/ SD (Pezeshki et al., 2020) | 63.41 | 7.77 |
| ResNet-56 w/ LfF (Nam et al., 2020) | 63.11 | 42.09 |

the network would exhibit a strong propensity to identify a picture that has a small black block on its top left corner as "apple", if it suffers from shortcut learning. We then designed a new testing scenario where we extracted all the testing samples from the other 99 classes(except apple) and then added a small black block on the same position on each of them (We then term this as **CIFAR99**). Since on CIFAR99, only the remaining 99 classes were modified. Once the network excessively relies on the decision rule that connects "images with a black block" with the class "apple", it would demonstrate a strong propensity to identify the samples on CIFAR99 with "apple", which would result in lower accuracy. Shortly speaking, the performance on CIFAR99 actually reveals how well the model could resist the propensity of shortcut learning.

Table 1 presents the performance of models which were trained on Shortcut-CIFAR100 and tested on both Shortcut-CIFAR100 and CIFAR99. As we can see, WP improves the model's generalization ability on Shortcut-CIFAR100 as usual. We want to highlight the huge improvement WP achieves on CIFAR99, where models without WP demonstrate a strong propensity to misidentify the images when these images exhibit similar patterns as those in other classes. To verify that WP is indeed learning to recognize more informative features, we visually plot the activation maps of all the models trained on Shortcut-CIFAR100. Figure 1 (b) provably demonstrates the effectiveness of WP in combating shortcut learning. After training on Shortcut-CIFAR100, the model without WP would drawn in the small block in the upper left corner while applying WP helps the model focus on more discriminative features. We also include spectral decoupling regularization (Pezeshki et al., 2020) and LfF (Nam et al., 2020) as comparisons. Both SD and LfF could improve the model's generalization ability (higher accuracy on Shortcuf-CIFAR100), but SD fails dramatically on getting over shortcut decision rule. [2] In short, these results of this conceptual experiment answer the first question positively and manifest the ability of WP to restrain the excessive reliance on dominant patterns when classifying.

---

[2]We hypothesis that such phenomenon would be relieved if we use the advanced variant of SD that imposes penalty separately for each class. But in this case(100 classes), it would entails a massive increase of hyperparameters (at least 100).

Regarding the *second question*, it is tempting to expect that there would be one or more ideal images that not only do not belong to the distribution of training data, but also are able to detect all the unintended dominant patterns. Alas, to precisely find such images require us knowing which patterns CNNs rely on, which is hard because patterns do not appear in a form amenable to discover ... so, not a viable option. Intriguingly, over all the alternative option, the solution with the white paper works best. As in Figure 2, four candidates were evaluated, namely "Gaussian Noise", "Ice-cream", "CIFAR-10", and "White Paper". We keep all the other implementation details unchanged and merely modified the images while training ResNet-56 on CIFAR-100. Specifically, "Gaussian Noise" experiments represent that we changed the white papers into images sampled from a standard normal distribution. "Ice-cream" denotes the whole ice-cream class of images from ImageNet while "CIFAR-10" denotes that all the images from CIFAR-10 were used for detection. Extensive details to facilitate replication are provided in the Appendix.C

Even with noise-generated images, there is still a performance boost over the vanilla model. Then with the increasing number of real-world images, the performances get higher. But white paper outperforms all the other solution. A possible explanation for this might be that the uninformative nature of the white paper seems to make it more suitable for detecting spurious dominant patterns, since the lack of semantics itself means no bias towards any pattern. Just like coloring on this white paper, the extent to which some pattern plays a dominant role for a class will be shown on the output distribution of the white paper. We also want to note that all the alternatives outperform the vanilla setting, indicating that the effectiveness of the whole detect-and-conquer practice.

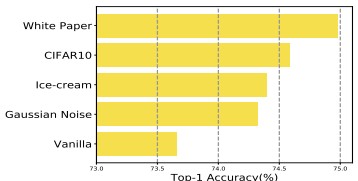

Figure 2: Effects of different types of images used in our scheme. The white paper outperforms the other solutions.

## 4 How does White Paper Assistance work?

After introducing our method, we then move on to a series of experiments used to glean insights on the behavior of the White Paper Assistance.

**What does the training with White Paper Assistance look like?** Analysis on the trend of training and testing accuracy is of vital importance to understand the effect of a method. Figure 3 (a) and (b) characterize the change of training and testing accuracy across epochs during training with and without WP. Note that we set $P = 1$, namely WP was steadily conducted after each epoch of real images training. Compared with its counterpart, training with WP exhibits a slower increasing trend on training accuracy, demonstrating that our approach helps suppress model from overusing shortcuts that could rapidly improve the generalization on training data otherwise. Even though the training error can both reach zero regardless of the use of our approach, training with WP achieves a significant performance boost on testing data, demonstrating better generalization ability. Not only that, the use of WP on the later stage of training can still provide further improvement with the model, as evident from the fact that training with WP achieves its best performance after epoch 225.[3]

It is still worth noting that after each time we conducted multiple iterations of white paper training, the testing accuracy would fall dramatically to around 1%. It is as if the model was guessing wildly at all the testing data. But when we moved on and fed real images, both the training and testing accuracy would restore and continue to rise (as seen from the continuous curves of both training and testing accuracy in Figure 3 (a) and (b)), as if the model was not affected by what just happened. Does the state of model performing random guess is a bad sign? Does this mean that White Paper Assistance is harmful? What happened with the model? Why would the accuracy could be restored? We will devote the next part to analyse the causes of it.

**Is white paper training harmful to the model?** The ultimate goal of training is to find better parameters of the model. To delve deeper into WP, we turn our attention to parameter changes.

---

[3]In this case, we decay the learning rate by factor 0.1 at epochs 150, 225. The training after epoch 225 often suffers from severe overfitting so that it fails to achieve further improvement.

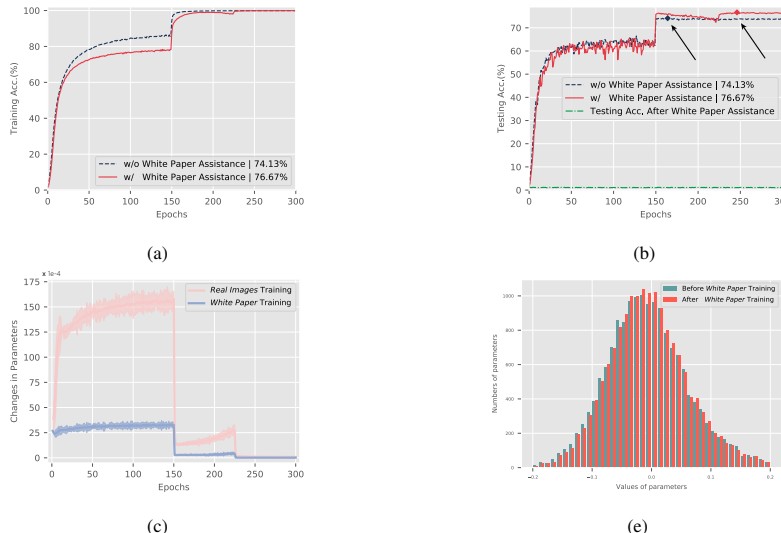

(a)

(b)

(c)

(e)

Figure 3: **Behavior of White Paper Assistance: (a, b)** The evolution of training/testing accuracy when training ResNet-110 with and without WP. **(c)** Changes in parameters of real images training and white paper training. We use L1 distance to measure the changes of parameters on the final convolutional layer of ResNet-110 when training WP with $P = 1$. **(d)** Parameter distributions before and after White Paper Assistance was conducted. This change happened on the final convolutional layer of ResNet-110 at epoch 100. More results of changes or distributions on other layers are present in Appendix.D.

First, we need to figure out which part of the model is more affected. A trained ResNet-56 $\mathcal{F}(\theta)$ that has achieved 73.51% accuracy on CIFAR-100 was picked. We use $\mathcal{C}$ and $f$ to denote the parameters of the projection head (*i.e.* all the convolutional layers) and the classification head (*i.e.* the last fully-connected layer) at this moment, respectively. Then, WP was applied on $\mathcal{F}(\theta)$ and we observed the performance dropping to 1%. Let $\mathcal{F}(\tilde{\theta})$, $\tilde{\mathcal{C}}$ and $\tilde{f}$ to denote the parameters of the whole network, the projection head and the classification head at this moment, respectively. To determine which part is more affected, we combined $\mathcal{C}$ with $\tilde{f}$ and combined $\tilde{\mathcal{C}}$ with $f$. As shown in Figure 4, for $\mathcal{F}(\theta)$, if we replaced its classification head, the accuracy changed little (73.51% → 73.4%), whereas the accuracy would drop dramatically (73.51% → 1%) when we replaced its projection head. These observations suggest that the modifications of WP mainly happen on the projection head, rather than the classification head. Similar conclusion could be drawn from $\mathcal{F}(\tilde{\theta})$.

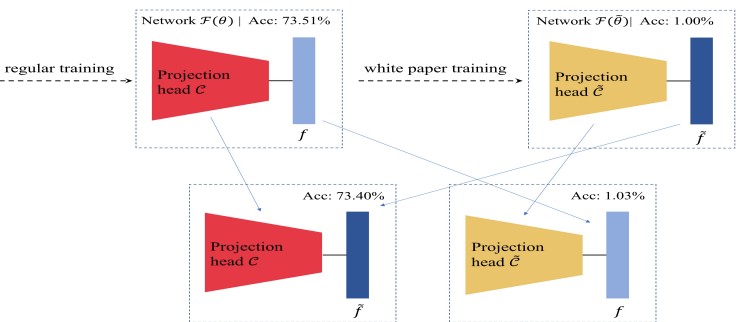

Figure 4: Illustration of how we determine which part is more affected by WP. We also extend this experiment to other models in Appendix.D.

Then we turn to quantitatively measure the changes of parameters due to our method. Since white paper training and real images training alternately appear when $P = 1$, we plot the changes of parameters using L1(mean absolute error) distance with respect to the epoch. In Figure 3 (c), we

observe that changes brought by white paper training are smaller compared to real images training. Figure 3 (d) depicts the distributions of the last convolutional layer's parameter of ResNet-110 before and after the white paper training at certain epoch. It can be seen that our approach is actually disturbing rather than devastating the distributions of parameters. We believe the fact that the model has not been completely devastated proves that White Paper Assistance does not damage the model, and could explain why the accuracy could be rapidly restored. In addition, these results are strong proof that CNNs are vulnerable – slight perturbations of parameters could bring down the whole generalization ability of the model, or at least it seems that way.

## 5 DOES WHITE PAPER ASSISTANCE PERFORM BETTER?

The section below shows the experimental results to prove that applying White Paper Assistance results in better performance. All experiments are implemented using Pytorch on 4×GTX1080Ti. If not specified, all experimental results reported are averaged over 4 runs. Being limited to the space, we supplement the implementation details in Appendix.E.

### 5.1 CLASSIFICATION

**White Paper Assistance performs better across different model architectures.** As discussed before, White Paper Assistance enjoys a "plug-and-play" property, namely it can be directly amendable to almost any CNNs without any changes needed on the network architecture. To prove it from an empirical standpoint, several different architectures were used to evaluate the effectiveness of our approach, namely, ResNet (He et al., 2016), SeNet (Hu et al., 2018), Wide ResNet (Zagoruyko & Komodakis, 2016), PyramidNet (Han et al., 2017), DenseNet (Huang et al., 2017), and ResNext (Xie et al., 2017). All the experiments were performed on CIFAR-100 (Krizhevsky, 2012), one of the most extensively studied classification benchmark tasks. In Table 2, we observe that White Paper Assistance consistently outperforms the baseline scheme. For example, our approach achieves 76.35% accuracy when applying Resnet-110 model, which is a 2% improvement over vanilla training. This wide applicability significantly promotes its value in practice and implies, once again, that learning shortcut cues is a common problem that CNNs cannot easily get rid of without special treatment like White Paper Assistance.

Table 2: Top-1 error rates(%) over different architectures.

| Model | w/o WP | w/ WP |
|---|---|---|
| Resnet-110 | 25.65 $\pm$ 0.07 | **23.65** $\pm$ 0.10 |
| Resnet-164 | 24.26 $\pm$ 0.06 | **22.89** $\pm$ 0.08 |
| SeResNet-110 | 23.36 $\pm$ 0.09 | **22.37** $\pm$ 0.04 |
| WRN-28-10 | 21.18 $\pm$ 0.01 | **19.80** $\pm$ 0.04 |
| DenseNet-100-12 | 22.78 $\pm$ 0.07 | **22.02** $\pm$ 0.02 |
| ResNext-29, 8×64 | 20.68 $\pm$ 0.16 | **19.41** $\pm$ 0.12 |
| PyramidNet-110-270 | 18.62 $\pm$ 0.03 | **17.96** $\pm$ 0.08 |
| PyramidNet-200-240 | 16.77 $\pm$ 0.07 | **16.13** $\pm$ 0.12 |

**White Paper Assistance gives better performance on various benchmarks.** We next performed experiments with different benchmark datasets. The following benchmark datasets were used: SVHN (Netzer et al., 2011), CIFAR-10 (Krizhevsky, 2012), tiny-ImageNet, CUB-200-2011 (Wah et al., 2011), StandfordDogs (Khosla et al., 2011), StandfordCars (Krause et al., 2013). As shown in Table 3, White Paper Assistance leads to better error rates in all the cases. Such wide applicability to datasets reveals that it is nearly impossible to create a shortcut-free dataset, which further proves the importance of avoiding reliance on unintended shortcuts.

**White Paper Assistance leads to further improvements over other techniques.** In practical use, it is common to use multiple techniques simultaneously in the hope of a higher boost on performance. Thus it imposes very high demands in term of compatibility for a technique. With this in

Table 3: Top-1 error rates(%) on different benchmark datasets.

| Dataset | Model | w/o WP | w/ WP |
|---------|-------|--------|-------|
| SVHN | ResNet-110 | 3.57 $\pm$ 0.02 | **3.28** $\pm$ 0.01 |
| CIFAR-10 | ResNet-110 | 5.61 $\pm$ 0.07 | **4.94** $\pm$ 0.04 |
| tiny-ImageNet | ResNet-110 | 37.66 $\pm$ 0.18 | **35.58** $\pm$ 0.05 |
| CUB | ResNet-50 | 32.36 $\pm$ 0.18 | **28.65** $\pm$ 0.05 |
| Standford Cars | ResNet-50 | 12.12 $\pm$ 0.02 | **9.87** $\pm$ 0.12 |
| Standford Dogs | ResNet-50 | 38.79 $\pm$ 0.74 | **34.55** $\pm$ 0.82 |

mind, we then applied White Paper Assistance with several widely used techniques, Mixup (Zhang et al., 2017), AutoAugment (Cubuk et al., 2018), FastAutoAugment(Lim et al., 2019), and Label Smoothing (Szegedy et al., 2016) into ResNet-110. The experiments presented here are designed to pinpoint the compatibility of WP. Consequently, we highlight that achieving state-of-the-art results is not the objective. As reported in Table 4, our approach consistently makes further improvements, which is reasonable as the training of white paper is independent of the training process of real images.

Table 4: Top-1 error rates(%) over different techniques.

| Method | vanilla | w/ WP |
|--------|---------|-------|
| + White Paper Assistance | 23.65 $\pm$ 0.09 | - |
| + Mixup | 22.13 $\pm$ 0.10 | **21.80** $\pm$ 0.23 |
| + AutoAugment | 21.70 $\pm$ 0.03 | **21.12** $\pm$ 0.29 |
| + FastAutoAugment | 22.46 $\pm$ 0.05 | **21.73** $\pm$ 0.15 |
| + Label Smoothing | 24.68 $\pm$ 0.07 | **23.41** $\pm$ 0.07 |

## 5.2 IMBALANCED CLASSIFICATION

The benchmark datasets we use above all exhibit roughly uniform distributions of class labels. But it is always prohibitively expensive to construct a real-world dataset with proper balance among classes, which explains the long-tailed label distributions that most real-world large-scale datasets (Cui et al., 2018; Gupta et al., 2019) have. On these datasets, a few dominant classes hold a large number of samples while a few other classes only possess relatively few samples. Conceivably, with the severe imbalance among the number of classes comes a severer imbalance among patterns, where White Paper Assistance can help. To verify our conjecture, we created the long-tailed version of CIFAR-10 and CIFAR-100 as (Cao et al., 2019), then validated the performance with and without our approach. We also include a combination of our approach with two other specialized approaches, CB-Focal (Cui et al., 2019) and LDAM (Cao et al., 2019). We strictly use the same parameter settings of (Cao et al., 2019).

We report the top-1 validation errors of various methods and combinations for long-tailed CIFAR-100 and CIFAR-10 in Table 5. We observe that White Paper Assistance alone can already improve over the vanilla setting, and the combinations of our approach with CB-Focal and LDAM achieve better performance gains, demonstrating both the versatility on imbalanced classification and the compatibility with other techniques. Overall, these observations strengthen the idea that White Paper Assistance is of great use to solve or alleviate this kind of pattern imbalance problems.

## 5.3 ROBUSTNESS

It is well known that the human vision system is not easily fooled by small changes in query images, whereas existing deep learning models may exhibit dramatic performance decline (Hendrycks et al., 2021). This proves again that the deep learning vision systems do not actually solve a task in the

Table 5: Top-1 error rates(%) of ResNet-32 on long-tailed CIFAR-10 and CIFAR-100. The imbalance ratio denotes the ratio between the numbers of samples of the most and least frequent classes.

| | Long-tailed CIFAR10 | | Long-tailed CIFAR100 | |
|---|---|---|---|---|
| Imbalance Ratio($N_{max}/N_{min}$) | 100 | 50 | 100 | 50 |
| CE(Baseline) | 28.71 $\pm$ 0.71 | 22.74 $\pm$ 0.05 | 61.24 $\pm$ 0.05 | 56.58 $\pm$ 0.09 |
| CE(Baseline) + WP | **27.68** $\pm$ 0.40 | **22.04** $\pm$ 0.18 | **60.15** $\pm$ 0.07 | **54.97** $\pm$ 0.11 |
| CB-Focal | 27.86 $\pm$ 0.16 | 23.34 $\pm$ 0.21 | 61.51 $\pm$ 0.20 | 56.28 $\pm$ 0.19 |
| CB-Focal + WP | **26.79** $\pm$ 0.07 | **22.26** $\pm$ 0.15 | **60.04** $\pm$ 0.04 | **55.11** $\pm$ 0.31 |
| LDAM-DRW | 22.59 $\pm$ 0.08 | 18.40 $\pm$ 0.15 | 57.43 $\pm$ 0.21 | 52.89 $\pm$ 0.51 |
| LDAM-DRW + WP | **21.15** $\pm$ 0.21 | **18.00** $\pm$ 0.23 | **55.13** $\pm$ 0.24 | **51.79** $\pm$ 0.09 |

way we intend them to, hence should be viewed as a symptom of models learning shortcuts. In order to check whether White Paper Assistance alleviates the shortcut learning indeed, we evaluate the robustness to common corruptions of our approach on tiny-ImageNet-C (Hendrycks & Dietterich, 2019). Specifically, we compare the classifiers' performance with and without WP across five corruption severity levels on each type of given corruption. Since each model performs differently on its own, each result was averaged by four runs. We have written the detailed calculations in the Appendix.F.

The results are shown in Table 6. As expected, White Paper Assistance clearly improves baseline's robustness against *all* the corruptions universally. These consistent improvements imply the potential of our approach to generalize to other untested types of corruptions. It's worth noting again that these improvements are achieved solely through the help of the white paper, which is uninformative and has no relationship with these noisy data. This suggests that our approach is indeed able to improve the robustness, and therefore implies that White Paper Assistance can have an effect on avoiding the excessive reliance of shortcuts.

Table 6: Corruption errors of tiny-ImageNet-C on different corruptions across five corruption severity levels. All metrics are top-1 error rates (for corrupted test sets, we average for 5-severity levels in four runs on ResNet-110. Separate results in each run are provided in Appendix.F.).

| | | NOISE | | | BLUR | | | | WEATHER | | | | DIGITAL | | | |
|---|---|---|---|---|---|---|---|---|---|---|---|---|---|---|---|---|
| Method | mCE | gaussian | shot | impulse | defocus | glass | motion | zoom | snow | frost | fog | brightness | contrast | elastic | pixelate | jpeg |
| w/o WP | 78.32 | 83.18 | 79.62 | 81.72 | 86.74 | 84.27 | 80.83 | 84.65 | 78.12 | 74.62 | 74.73 | 69.86 | 88.71 | 75.81 | 64.44 | 67.54 |
| w/ WP | **75.87** | **82.12** | **77.83** | **80.44** | **84.29** | **82.77** | **78.08** | **81.87** | **75.31** | **72.00** | **72.23** | **65.98** | **87.32** | **72.82** | **61.29** | **63.80** |

# 6 CONCLUSION

In this paper, we propose an interesting method, called White Paper Assistance, used to alleviate the excessive reliance of models on shortcut learning. Inspired by the common sense of the printer test page, our framework utilizes the white paper to detect the unintended propensity for dominant features that lack of generalization utility and then "debiases" the model by enforcing the model to produce a uniform output distribution on the white paper. By adding synthetic shortcuts to the training data, we ensure that our approach can help remove shortcut features from the data and improve the generalization ability. Besides, through extensive experiments, our method shows successful results along three axes: classification, imbalanced classification and robustness. The wide applicability, compatibility and versatility highlight the progress of our method in overcoming shortcut learning, which thus should be viewed as *a step forward towards a shortcut-free training*, as we claimed in the title.

We hope this study could facilitate the awareness for shortcut learning, and more importantly, open up promising avenues towards fair, robust, trustworthy deep learning.

## 7 REPRODUCIBILITY STATEMENT

To ensure the reproducibility of our work, we provide a pytorch working example source code of our method in supplementary materials. We will make this project open-source after the whole review process. We also provide a implementation code of Shortcut-CIFAR100 and CIFAR99 if you are interested.

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

## A    MORE RESULTS ON CONCEPTUAL EXPERIMENTS

### A.1    MORE CAM VISUALIZATIONS

For better understanding, we plot more examples on cam visualization here.

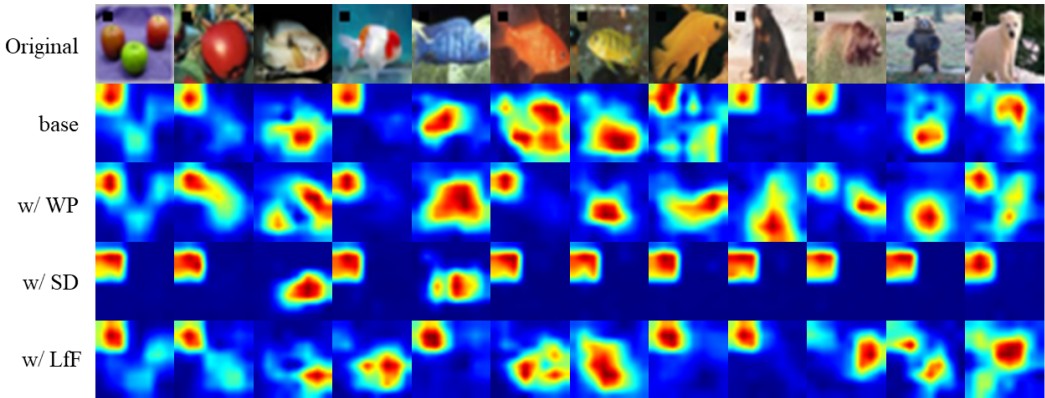

Figure 5: Supplementary cam visualizations on models trained on Shortcut-CIFAR100.

### A.2    MORE EXPERIMENTS RESULTS

In the original experiments, we modified the first class in Shortcut-CIFAR100 and modified the other 99 class testing samples in CIFAR99. We extend this experiments by modifying the third class(fish). Specifically, in the revised Shortcut-CIFAR100, only samples of the third class are modified. In the revised CIFAR99, we held out the testing sample of the third class and revise all the testing samples of the remaining 99 classes. The results are presented in the following table.

Table 7: Top-1 accuracy(%) on revised Shortcut-CIFAR100 and revised CIFAR99.

| Model | revised Shortcut-CIFAR100 | revised CIFAR99 |
|---|---|---|
| ResNet-56 | 63.57 | 25.55 |
| ResNet-56 w/ WP | **66.75** | **29.77** |
| ResNet-56 w/ SD (Pezeshki et al., 2020) | 62.76 | 6.69 |
| ResNet-56 w/ LfF (Nam et al., 2020) | 63.29 | 27.57 |

## B    MORE QUESTIONS ON DESIGNING WHITE PAPER ASSISTANCE

Q3.    Why repeat the process for multiple iterations?

Regarding this question, we answer it with intuitive reasoning and empirical evidence. Intuitively, a biased model cannot be perfectly amended with ease. Take "polishing" as a supportive example. The removal of unintended oxidization requires polishing on the appearance of an item back and forth, until a smoothing finish is accomplished. To verify the intuition, we conduct a controlled trial that we kept $P = 1$ unchanged while modified $M$ from 50 to 500. Aligned with our expectation, more rounds of schemes lead to a better refinement in learning, which results in a better generalization performance. Usually, we find that the performances often get the highest when $M$ is approximately equal to the times real training images are trained per epoch, namely when the amount of training on the white paper is compatible with the amount of training on the real images.

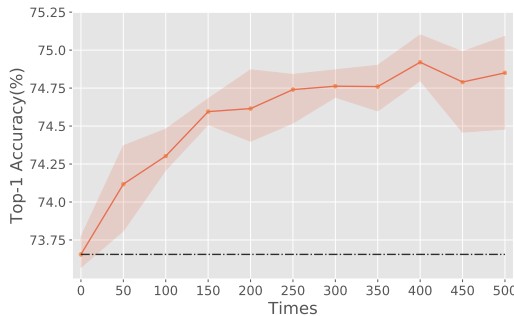

Figure 6: Final test accuracy as a function of iterations that we repeat the process for. The shaded areas represent the minimum and maximum results from 4 runs. Dashed dotted line denotes the baseline accuracy. Our approach receives near optimal performances when the amounts of training on white paper and real images are close to each other(In this case, the times of iteration of real image training for one epoch are 391).

## C    EXPLANATIONS ON DIFFERENT TYPES OF IMAGES

Here we provide some further explanations on experiments regarding why choose using the white paper. In these experiments, we kept all the scheme details unchanged but only modified the images fed into the model(as in Algorithm 1, line 7).

**Vanilla** The baseline settings of training ResNet-56 on CIFAR-100.
**Gaussian Noise** Images randomly sampled from standard normal distribution.
**Ice-cream** Ice-cream images randomly sampled from ImageNet(Ice-cream class, n07614500).
**CIFAR-10** Images sampled from CIFAR-10.

## D    MORE EXPERIMENTS EXPLORING BEHAVIOR OF WHITE PAPER
    ASSISTANCE

To begin with, we extended the experiments in Figure 4 (that determine the part that White Paper Assistance has more effect on) to other models. Two models were used to further demonstrate that White Paper Assistance mainly modifies the projection head and has little effect on the classification head of models(Table 8).

Table 8: Results on other architectures to suggest that the modifications of White Paper Assistance mainly happens on the convolutional layers.

| Model | Original $\mathcal{C} + f$ | After White Paper Training $\tilde{\mathcal{C}} + \tilde{f}$ | Combination #1 $\mathcal{C} + \tilde{f}$ | Combination #2 $\tilde{\mathcal{C}} + f$ |
|---|---|---|---|---|
| ResNet-110 | 74.18% | 1.13% | 74.26% | 1.10% |
| Pyramid-110-270 | 81.31% | 1.00% | 81.30% | 1.00% |

In Section 5, we still report that White Paper Assistance would incur smaller changes in parameters and such changes would not devastate the inhere parameter distributions. The observations happen on the last convolutional layer of ResNet-110. Here we present more results about the changes and parameter distributions on other layers in Figure 7.

## E    IMPLEMENTATION DETAILS

We describe the training implementation settings in detail.

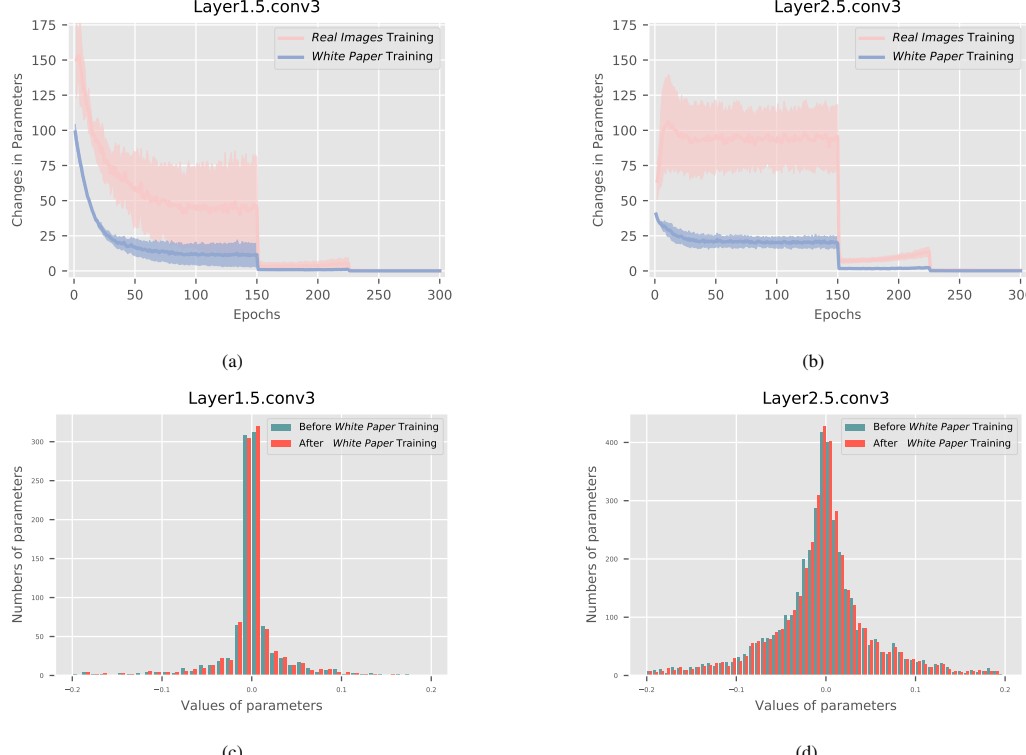

Figure 7: **Behavior of White Paper Assistance on other convolutional layer (a,b)** Changes in parameters of real images training and white paper training on other convolutional layers. The shaded areas indicate the standard deviations across 4 independent trails. **(c,d)** Parameter distributions before and after White Paper Assistance conducted on other convolutional layers of ResNet-110 at epoch 100.

### E.1 CLASSIFICATION

**CIFAR-100** For ResNet, SeResNet and ResNext, we set the number of training epochs to 300. The learning rate were set to 0.1 and was decaying by the factor of 0.1 at epoch 150 and 225. We used SGD optimizer, and the minibatch size, momentum, weight decay were set to 128, 0.9, and 0.0001, respectively. When training, we set $P = 1, \lambda = 1$.

For Wide ResNet, we changed the training epochs to 200. Then the learning rate was decaying by the factor of 0.2 at epoch 60, 120, 160, respectively. When training, we set $P = 1, \lambda = 0.5$.

For DenseNet, we changed the mini-batch size to 64 following the common practice. When training, we set $P = 1, \lambda = 1$.

For PyramidNet, the learning rate rose to 0.25 and we set $P = 1, \lambda = 0.5$ when training.

For all the experiments on CIFAR-100, we adopted the standard data augmentation techniques including Horizontal Flipping and Random Cropping.

**CIFAR-10** We adopted the same training strategy as on CIFAR100.

**SVHN** We held all the hyper-parameters unchanged but did not adopt any data augmentation techniques. When training, we set $P = 1, \lambda = 0.1$.

**tiny-ImageNet** We adopted the same training strategy as on CIFAR100 except that we changed the mini-batch size to 256.

On fine-grained benchmark datasets, we trained ResNet-50 from scratch and set the training epochs to 300 to ensure convergence. The learning rate was set to 0.1 and decayed by 0.1 at epoch 150, 225. The mini-batch size, momentum, and weight decay were set to 16, 0.9, and 0.0001, respectively. The following contents move on to discuss the data augmentation techniques we used.

**CUB-200-2011 and Standford Cars** For these two datasets, we first resized images to $600 \times 600$ and cropped them to $448 \times 448$. Then we adopted the Random Horizontal Flipping.

**Standford Dogs** For the Standford Dogs, we resized images to $256 \times 256$ and cropped them to $224 \times 224$. Then we also adopted the Random Horizontal Flipping.

**Combinations on CIFAR-100** When combined with Mixup, we set $P = 1, \lambda = 1$. For Autoaugment, we adopted the implementation of publicly available code.[4]. When training, we set $P = 1, \lambda = 1.5$. For FastAutoaugment, we adopted the implementation of publicly available code [5] and set $P = 1, \lambda = 1$. For label smooth, we set the smoothing parameter to 0.1 and set $P = 1, \lambda = 1$.

### E.2    IMBALANCED CLASSIFICATION

Both Long-tailed CIFAR-100 and CIFAR-10 follow an exponential decay in sample size across different clasess. The ratio $\rho$ is used to denote the ratio between sample sizes of the most frequent and least frequent class. Figure 8 depicts the sample distributions of Long-tailed CIFAR-100.

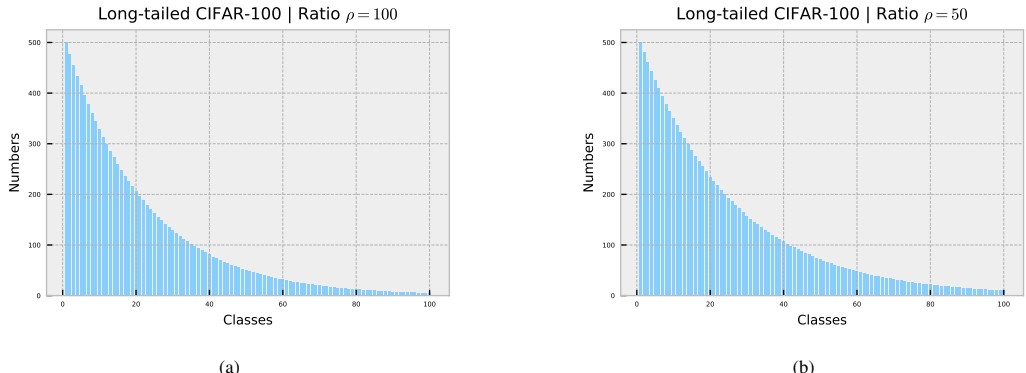

(a)                                                                          (b)

Figure 8: **Number of training examples per class in Long-tailed CIFAR-100.**

For implementation details, we adopted the same setting as those suggested in (Cao et al., 2019).[6]

## F    METRICS AND COMPREHENSIVE RESULTS ON ROBUSTNESS

To evaluate the performance of White Paper Assistance on robustness to common corruptions, we refer to the metrics used in tiny-ImageNet-C. Specifically, when evaluating performances of baselines, we took four trained classifiers $f$ trained with vanilla settings. Then we tested the classifier on each corruption type $c$ at each level of severity $s \in \{1, 2, 3, 4, 5\}$. We used $E_{s,c}^f$ to denote the top-1 error rates. Then the average corruption error of all baseline models in corruption $c$ should be:

$$CE_c = \frac{1}{4 \times 5} \sum_{f=1}^{4} \sum_{s=1}^{5} E_{s,c}^f \tag{2}$$

Then we aggregated all the $CE_c$ to compute the mean corruption error values to all the corruption:

$$mCE = \frac{1}{15} \sum_{c=1}^{15} CE_c = \frac{1}{4 \times 5 \times 15} \sum_{c=1}^{15} \sum_{f=1}^{4} \sum_{s=1}^{5} E_{s,c}^f \tag{3}$$

---

[4]`https://github.com/DeepVoltaire/AutoAugment/blob/master/autoaugment.py`
[5]`https://github.com/ildoonet/cutmix/blob/master/autoaug/archive.py`
[6]`https://github.com/kaidic/LDAM-DRW`

Here, we report the results of all the models tested in Table 9.

Table 9: Comprehensive corruption error results of baseline and White Paper Assistance on single model. "Clean" here denote the top-1 error rate of the model on the clean version of tiny-ImageNet.

| Model | Clean | avgCE | NOISE | | | BLUR | | | | WEATHER | | | | DIGITAL | | | |
|---|---|---|---|---|---|---|---|---|---|---|---|---|---|---|---|---|---|
| | | | gaussian | shot | impulse | defocus | glass | motion | zoom | snow | frost | fog | brightness | contrast | elastic | pixelate | jpeg |
| ResNet-110 | 37.41 | 77.29 | 82.03 | 77.89 | 81.07 | 86.28 | 83.91 | 79.92 | 83.91 | 77.05 | 72.83 | 73.29 | 69.23 | 87.70 | 75.32 | 62.31 | 66.68 |
| ResNet-110 | 37.21 | 78.41 | 84.16 | 80.99 | 81.85 | 85.84 | 83.97 | 80.27 | 83.56 | 78.72 | 75.51 | 75.16 | 69.13 | 88.97 | 75.35 | 64.96 | 67.75 |
| ResNet-110 | 37.90 | 78.76 | 83.01 | 79.60 | 81.85 | 87.18 | 84.29 | 81.52 | 85.53 | 78.51 | 74.85 | 74.07 | 70.95 | 88.56 | 76.53 | 66.37 | 68.63 |
| ResNet-110 | 38.11 | 78.81 | 83.50 | 79.98 | 82.10 | 87.67 | 84.90 | 81.60 | 85.60 | 78.18 | 75.28 | 76.38 | 70.14 | 89.60 | 76.05 | 64.11 | 67.08 |
| ResNet-110+WP | **35.39** | 76.01 | **81.08** | **76.86** | 80.00 | 85.16 | 82.78 | 79.41 | 82.90 | 74.85 | 72.31 | 73.09 | 66.74 | 87.63 | 73.14 | **60.73** | 63.44 |
| ResNet-110+WP | 35.60 | 75.66 | 82.84 | 78.58 | 80.38 | **83.34** | 82.55 | **76.86** | **80.69** | 75.37 | **71.43** | 71.92 | 66.05 | 87.65 | 72.45 | 61.12 | 63.70 |
| ResNet-110+WP | 35.76 | 76.28 | 82.77 | 78.08 | 81.56 | 84.73 | 83.24 | 78.36 | 82.06 | 76.17 | 72.73 | 72.55 | 65.57 | 87.12 | 72.32 | 61.27 | 64.64 |
| ResNet-110+WP | 35.56 | **75.55** | 81.80 | 77.78 | **79.81** | 83.91 | **82.50** | 77.67 | 81.84 | **74.84** | 71.54 | **71.34** | **65.54** | **86.88** | **72.38** | 62.03 | **63.41** |

## G   EFFECTS OF PARAMETERS

We evaluated the effect of probability $P$ and strength $\lambda$ when training ResNet-56 in CIFAR-100 using the aforementioned settings. We first inspected the performances for different choices of $P \in \{0, 0.2, 0.4, 0.6, 0.8, 1\}$ when $\lambda = 1$. Specifically, $P = 0$ denotes the baseline without applying White Paper Assistance. In Figure 9 (a), we observe that White Paper Assistance consistently achieves a performance boost even in a small participation rate. Then we turned our attention to another hyper-parameter $\lambda$ that also plays an important role during training. Here we tried different choices with $\lambda \in \{0, 0.2, 0.4, 0.6, 0.8, 1.0, 1.2, 1.4, 1.6, 1.8\}$ while keeping $P = 1$. Figure 9 (b) characterizes the evolution of performances on varying $\lambda$. The best performance can be achieved when $\lambda$ is set to 1.

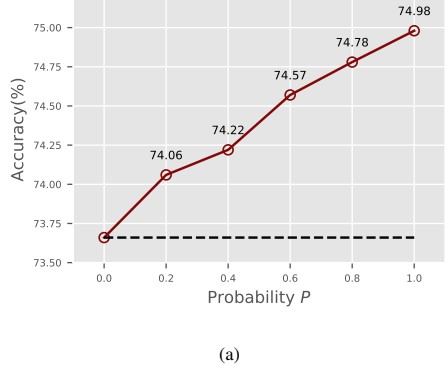

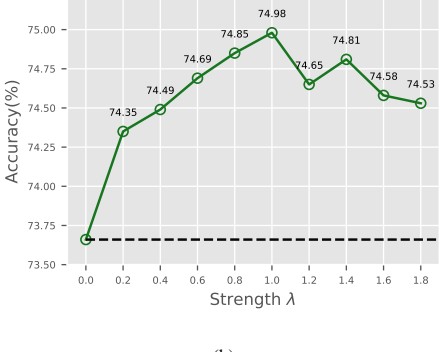

(a)                                           (b)

Figure 9: **Effects of Probability $P$ and Strength $\lambda$.**

