# OpenReview forum: "White Paper Assistance: A Step Forward Beyond the Shortcut Learning"
_ICLR.cc/2022/Conference — ICLR 2022 Submitted_

### Official Review · Reviewer_C42n · 2021-10-25

**Correctness:** 3
**Technical Novelty And Significance:** 2
**Empirical Novelty And Significance:** 2
**Recommendation:** 3
**Confidence:** 3

**Main Review:**

Strengths:
-  The proposed method is intuive, simple to implement, and very general.
-  The problem of preventing "shortcuts" is an important one.

Weaknesses:
-  The experiments have a fundamental flaw that prevents the results from being meaningful:  the results reported are the "best test accuracy achieved across the entire training process" instead of the standard "test accuracy of the model with the best validation performance across the training process."  This conclusion is based on Figure 3 and an inspection of the code.
	-  This might be related to the conflict between the stated intuition that "predicting uniform probability for a white image should help the model" and the empirical results that "predicting uniform probability for real images (eg, CIFAR images) also helps the model."
-  The connection to the "shortcut" literature should be expanded upon.
	-  The related work is missing many references which include, but are not limited to: Shetty et al "Not Using the Car to See the Sidewalk--Quantifying and Controlling the Effects of Context in Classification and Segmentation";  Rieger et al "Interpretations are useful: penalizing explanations to align neural networks with prior knowledge";  Teney et al "Learning what makes a difference from counterfactual examples and gradient supervision"; Singh et al "Don't Judge an Object by Its Context: Learning to Overcome Contextual Bias"
	-  It is unclear what shortcuts this work detects, which is something that many methods in the "shortcut" literature specify.
	-  The evaluation is run on a testing distribution which matches the training distribution, which is unsusual for methods in the "shortcut" literature (as using the "shortcut" usually makes it easy to acheieve high accuracy on the training distribution, so showing improvements in this way by preventing "shortcuts" is very challenging).
	-  While setup correctly, the example using "black squares" and the CIFAR dataset is a very simplistic (and artificial) example of a "shortcut".  See the aforementioned references for some other examples.
	-  Generally, it seems that the proposed method is closer to a standard regularization technique than a method for preventing "shortcuts".



**Summary Of The Paper:**

The work proposes a method for reducing the extent to which a model learns to rely on "shortcuts" by introducing a new form of regularization encourages the model to make uniform predictions for white images. The proposed method is evaluated across a range of model architectures and datasets and evaluated in combination with a variety of existing techniques.


**Summary Of The Review:**

On its own, the methodology used to gather the results disqualifies the paper for publication at this point.  If the reviewer is mistaken about this, then all of the sub-scores would increase.

However, even if the reviewer is mistaken about this methodological error and the results are valid, it does not seem that the paper is ready for publication because the connection to the "shortcut" literature needs clarification (which impacts how almost the entire paper is written).

# Update based on author responses

The author's responses have addressed some of my concerns (eg, using testing data as a validation set, the lack of overlap between CIFAR10/100).  However, the paper still needs to be revised to emphasize that it is not focused on "perceptible patterns" as shortcuts (eg, cows are usually in pastures) but rather on "imperceptible patterns" (eg, high frequency signals in an image).  So I've changed my score from a 1 to a 3.

I'm very familiar with the work on "perceptible patterns" and this paper is not up to par for that area because  it does not identify specific shortcuts and then demonstrate that it has reduced them.  I'm much less familiar with the work on "imperceptible patterns".  So I've changed my confidence from a 5 to a 3.

---

> ### Author Response · Authors · 2021-11-17
> **To Reviewer C42n**
>
> Thanks anyway for your time and effort. Below please find our responses to your concerns point by point.
>
> 1.**Missing Validation Performance** There seems to be a fundamental misunderstanding here. We have to clarify that nearly all the datasets we used(except for tiny-ImageNet) do not provide a validation set. It is actually common sense [1,2,3,4] for these datasets to train with all the training data. All the policies and hyperparameters we used are fair and open(all the implementation details are in Appendix.E), following the common practice. We did not include any design that we could potentially benefit from. More importantly, all the comparisons between WP and the counterparts were made under the same circumstance. Therefore, the results in this paper are meaningful.
>
> 2.**Conflict between the stated intuition and the empirical results.** We respectfully disagree with you about the conflict. One of the key ideas of the whole scheme is that as long as the input images (whether they are white papers or others) do not belong to any classes the model has learned, a trained model should give an inference result that is as if it makes a random guess (page 3). Therefore, any images that could apply this logic would actually have an effect in detecting shortcuts. The improvements with WP-realimages not only verify this point emprically, but also prove the effectiveness of the detect-and-conquer scheme.
>
> 3.**Missing related work** The works you list are indeed related. We will cite them accordingly in our revised manuscript.
>
> 4.**What shortcuts this work detects** We cannot fully agree with this comment. Unlike other areas that are based on prior knowledge, shortcut learning refers to the propensity that the model relies more on statistically dominant features in the data. Such propensity can be caused both by the model and by the the dataset. Therefore, as we discussed in our paper (page 1),
> >Most patterns that CNNs rely on to classify do not appear in a form amenable to discover.
>
> This notion also appears in other explanations, such as in Section 4.1 of [5]:
> > Deep learning has led to the discovery of *much more subtle shortcut features*, including high-frequency patterns that are almost invisible to the human eye.
>
> also in [6]
> > Spurious correlations *do not appear to be stable properties*. Most datasets are not provided in a form amenable to discover stable properties.
>
> Therefore, considering the unstable properities of shortcut, the wide applicability of WP on various datasets manifests the generalization of WP to a variety of potential shortcuts, which should be viewed as an advantage.
>
> 5.**Evaluation on an i.i.d testing distribution** We would like to clarify that we have indeed included a testing scenario that is different from the distribution of the training data, which is the CIFAR99. The distribution of CIFAR99 is totally different from the training distribution of Shortcut-CIFAR100. In the latter, only one class has black squares while in the former, all the other classes have black squares. Thus as a conceptual experiment, CIFAR99 could reveal the extent of which the model could resist the propensity of shortcut, where WP has achieved superior results.
>
> Regarding the other evaluation, as we have discussed in Response4, it is unrealistic to design a corresponding testing scenario for each dataset since one might not know in advance which shortcuts exist. Besides, the improvement on top of the baseline itself represents the increased generalization, thus reflects the ability to combat shortcuts to some extent.
>
> 6.**Black squares being simple** Though being simplistic, this setting has devastated much of the perception of the model. When adding a black square, there is only a 40% probability that the data would be correctly classified. Therefore, the huge improvements on top of this should not be ignored. Besides, the CIFAR dataset is arguably the most commonly used dataset in the computer vision. It is more complex, and importantly, closer to many industrial application scenarios.
>
> 7.**Closer to a regularization technique** We respectfully disagree with you on that. We believe that being a regularization technique is not contradictory with being a method for shortcut. In fact, lots of regularization techniques have positive effects on alleviating shortcuts. Our method is designed to alleviate the shortcuts, and then a more robust decision rule leads to improved performance. This logic itself is coherent and reasonable. Therefore, all the performance boosts represent the wide applicability, which should be viewed as an advantage instead of a weakness.
>
> [1] Deep residual learning for image recognition. CVPR2016
> [2] Deep networks with stochastic depth. ECCV2016
> [3] Deep pyramidal residual networks. CVPR2017
> [4] Deep networks with stochastic depth. ICCV2019
> [5] Shortcut Learning in Deep Neural Networks. Nature Machine Intelligence 2020
> [6] Invariant Risk Minimization. 2019

---

> > ### Comment · Reviewer_C42n · 2021-11-17
> > **Thank you for these clarifications.  I had not realized that the focus was on "imperceptible patterns" that the model may be using as shortcuts.**
> >
> > 1.  Regardless of whether or not this is common practice, it is possible to overfit to the validation set (in this case, the testing set) and I would much rather see a small portion of the training data split off to use for validation (even if the final results get worse).  However, since this may be common practice, I'll leave it up to the AC to decide whether or not this is a serious concern.
> >
> > 2.  Thank you for pushing back on this.  I had not realized that there is very little to no overlap between the classes of CIFAR10/100.
> >
> > 4.  I'd recommend adjusting (at least) the Abstract and Introduction to make it clear that "shortcut learning" refers to "imperceptible patterns" because these sections make sense with both this definition and the alternative definition of "perceptible patterns".  Relatedly, the Related Work could be clearer by explicitly separating the discussion into paragraphs about perceptible/imperceptible patterns.
> >
> > 5.  This comment was about the results in Figure 2 and made under the assumption that the paper was about "perceptible patterns".  If the paper is revised to clarify that its focus is on "imperceptible patterns", then it doesn't really apply.
> >
> > 6.  While CIFAR is a challenging and realistic dataset, a version of it with a black square in the same location of every image of a single class is not.  Also, this "black square" is clearly a "perceptible pattern."
> >
> > 7.  This comment was also made under the assumption that the paper was about "perceptible patterns."  If the paper is revised to clarify that its focus is on "imperceptible patterns", then it doesn't really apply.

---

> > > ### Author Response · Authors · 2021-11-18
> > > **Further Responses**
> > >
> > > Thank you for taking the time to give a further reply.
> > >
> > > **On overfitting** It seems that the misunderstanding still exists. We would like to further clarify that the overfitting generally represents the phenomenon that the model fits the training data too well (*overfit in the training data*) whereas fails to fit additional data (in this case, testing dataset). Put another way, the overfitting generally leads to a performance drop in the validation/testing dataset rather than a performance increase. Therefore, the increase brought by WP is a production of the success against overfitting, rather than the sign of WP suffering from overfitting. Similar logic can be found in almost all the articles about regularization[1,2,3] or data augmentation[4,5].
> > >
> > > **Imperceptible pattern v.s. perceptible pattern**
> > > - First, what we describe shortcut in this paper did not restrict it to be "perceptible" or "imperceptible". In our paper, we noted
> > > >The neural network is biased towards capturing statistically dominant features in the data so that it starves the learning of other very informative but less frequent features.
> > >
> > >   This can also be applied to perceptible patterns. We take your recommended paper [6] as an example. In that paper, authors describe "Contextual bias" as
> > > >as b almost always co-occurs with c, the network may learn to inadvertently rely on pixels corresponding to c to predict b.
> > >
> > >   Clearly, in this case, the dominant feature is c and the less frequent feature is b. In other words, the shortcut we describe itself contains perceptible ones and imperceptible ones.
> > >
> > > - Second, the so-called "perceptible patterns" essentially stem from human judgment. For the perception of the model, there is no difference between the perceptible and the imperceptible patterns. It only matters when we design a solution. The perceptible patterns are based on a fundamental premise: a human must go through the datasets to statistically discover potential risk. In fact, we indeed describe a similar idea in page 1
> > > >It seems that the most reasonable and direct way is to identify which features contain shortcuts (like green to frogs) and which features should be enhanced (like shapes to animals)?
> > >
> > >   We believe such operations often entail
> > > >specific expert knowledge, let alone extensive manpower and resources.
> > >
> > >   On the contrary, partial superiority of WP lies in the fact that it does not require supervision/experience from human.
> > >
> > > - Third, since the distinguishing "perceptible and imperceptible" relys on the subjective congntion. Such taxonomy may be sloppy due to (1) ambiguous boundary (2) the congntion may vary from person to person.
> > >
> > > Therefore, we disagree with you that we must include such discussion which was redundant and loose. However, considering there is indeed an active line of research that focus on problems induced by specific bias(contextual bias, age bias), we will still cite these papers (including those you recommended) accordingly but with a cautious category name(for example, specific bias).
> > >
> > > **Black square** The black square indeed can be viewed as a "perceptible" pattern. But, we were doing this because we needed to concretize the shortcut into a special scenario where we could quantify it in numbers. The experiment is just a "proof of concept", rather than an ultimate application scenario. But the results did show that WP has the ability of alleviating the specific bias that we randomly designed.
> > >
> > > The ultimate application scenarios are actually the following experiments where we would like to solve this problem in scenarios closer to the real-world industrial application. These standard benchmarks pinpoint the efficacy of our method to alleviate shortcuts, and more importantly, the benefits when we alleviate them. These improvements are able to suggest the effectiveness of WP on "imperceptible" patterns, since we did not include any prior knowledge through any experiments. Therefore, the key idea of this paper that WP can alleviate shortcuts is well-established.
> > >
> > > [1] Deep networks with stochastic depth. ECCV2016
> > > [2] Can We Gain More from Orthogonality Regularizations in Training Deep CNNs? NIPS2018
> > > [3] DropBlock: A regularization method for convolutional networks. NIPS2018
> > > [4] CutMix: Regularization Strategy to Train Strong Classifiers with Localizable Features. ICCV2019
> > > [5] KeepAugment: A Simple Information-Preserving Data Augmentation Approach. CVPR2021
> > > [6] Don't Judge an Object by Its Context: Learning to Overcome Contextual Bias, CVPR2020

---

### Official Review · Reviewer_y5Su · 2021-11-01

**Correctness:** 1
**Technical Novelty And Significance:** 1
**Empirical Novelty And Significance:** 1
**Recommendation:** 1
**Confidence:** 5

**Main Review:**

The paper tackles and illustrates an important issue of not only CNN's but general deep learning models. As the authors describe this is not a novel insight and several approaches were introduced to reduce and even prevent shortcut learning by deep neural networks. However, I major line of related work seems to be missing: Teso et al. Explanatory Interactive Machine Learning (AIES 2019), Ross et al. Right for the Right Reasons (IJCAI 2017), Schramowski et al. Making deep neural networks right for the right scientific reasons by interacting with their explanations (Nat Mach Intell 2020), to name a few.

More importantly, the paper is not well written and even looks unfinished in parts (e.g first paragraph on page 5). This makes it very hard to read.

Since, in the vision domain, shortcut learning is already a well-explored field (see above) benchmark datasets were already introduced (Decoy MNIST, ColorMNIST, CLEVR-Hans) cf e.g. https://arxiv.org/abs/1703.03717, https://arxiv.org/abs/2011.12854 and https://www.nature.com/articles/s41467-019-08987-4.

Instead of designing or synthetic adapting datasets, I recommend an evaluation of already introduced and well-known datasets. Further, a comparison to previous approaches is missing in the present work.

**Summary Of The Paper:**

The present work introduces an approach to tackle Shortcut Learning by CNNs, called White Paper Assistance (WP). After motivating and introducing the method, the authors evaluate it on computer vision datasets with synthetic inserted shortcuts, ie. black pixels in the corners of the images. The authors show that the WP approach reduces the learning of so-called shortcuts.

**Summary Of The Review:**

Unfortunately, I do not believe that this paper meets the standard for publication. Besides that, the paper is not well written and therefore hard to understand its experimental evaluation is missing a comparison to previous approaches.

---

> ### Author Response · Authors · 2021-11-17
> **To Reviewer y5Su**
>
> Thanks anyway for your time and effort. Below please find our responses to your concerns point by point.
>
> 1.**Missing Related Work** The related works you list are helpful. We will cite them accordingly in our revised manuscript.
>
> 2.**Writings** The first paragraph on page 5 is actually a finished version. In this paragraph, we tried to answer the reason that we use the white paper instead of other inputs in terms of experimental results. If you have found any grammatical or expression errors, we would accept them in good faith and correct them in the manuscript. Considering that the Reviewer w6iS
> >"The paper is very well-written, especially the first three sections"
>
> and the Reviewer u8Nt
> >"The paper is easy to read and understand, and the arguments are mostly clear and concise."
>
> >"I enjoyed reading this paper"
>
> have all agreed that our manuscript is well-written, we believe our writings have achieved the effect we want.
>
> 3.**New dataset design**
>
> - First, the Shortcut-CIFAR100 used in this paper represents a conceptual shortcut scenario that model may exhibit a strong propensity to identify a picture that has a small black block when it has learned enough synthetic examples. The error rate in CIFAR99 would be a perfect indicator to detect this propensity. In fact, the discrepancy between the performance on Shortcut-CIFAR100 and CIFAR99 manifests the seriousness and harm of such shortcut. Consequently, the improvement WP achieves in this experiment indeed shows the ability of WP to restrain the excessive reliance on dominant patterns. We also include CAM visualization as further proof for better understanding. Therefore, we believe this performance boost is effective, meaningful and persuasive.
>
> - Second, the selection of the dataset is not arbitrary. The CIFAR dataset is arguably the most commonly used dataset in the computer vision domain. Compared with the MNIST dataset, CIFAR-100 is more complex, and more importantly, closer to the industrial application scenarios (*natural objects vs. digits; colored vs. black; 100 classes vs. 10 classes*). We did not include any design that we could potentially benefit from. We have even added more visualization experiments(Appendix A.1) and extended experimental results on other classes(Appendix A.2) to avoid occasionality. If you find any flaws, please let us know.
>
> - Third, the proposal of a new dataset or a new evaluation criterion is fairly common. Take your recommended article as an example, in this paper [Making deep neural networks right for the right scientific reasons by interacting with their explanations, 2020], authors present a new dataset(*Decoy Fashion-MNIST*) as an evaluation without validating his method on the old dataset(for example *colored MNIST*).
>
> - Fourth, the prosperity of a field requires a constant influx of new things, including new methods, new datasets, new criteria. The progress of the community does not necessarily rely on the review of past standards. Currently, the research on shortcut learning is still fragmented into various communities. We believe this work has the ability to fuel discussions across these different communities and initiate a movement that pushes for a new criterion used for shortcut evaluation. Hopefully, you could be open-minded for our proposed work.
>
> 4.**Missing comparison** We would like to clarify that we have performed comparison along multiple dimensions. On shortcut alleviating, we compared our work with Gradient Starvation[Gradient Starvation: A Learning Proclivity in Neural Networks, NIPS2021] and LfF[Learning from Failure:Training Debiased Classifier from Biased Classifier, NIPS2020] in page 4. On classification, we made comparison with several widely used techniques, including Mixup, AutoAugment, Fast AutoAugment, and label smoothing. On Imbalanced Classification, we compared with CB[Class-balanced loss based on effective number of samples, CVPR2019] and LDAM[Learning imbalanced datasets with label-distribution-aware margin loss, NIPS2019].

---

> > ### Comment · Reviewer_y5Su · 2021-11-18
> > **Still unclear motivation and contribution**
> >
> > Thank you for the clarifications. However, I still do not understand why the authors do not show the performance of the proposed approach on already investigated shortcut problems. Especially if the authors aim to investigate scenarios close to industrial application.
> >
> > Shortcut learning is not a crucial problem in CV benchmarks, however, it often prevents the applications of deep learning algorithms in industrial settings. Don’t get me wrong, I agree that it is a good first step to evaluate on benchmarks. But also, in this case, several synthetic datasets already exists. For instance, popular CV benchmark datasets with know shortcuts exits, e.g. Pascal VOC 2007 presented in https://www.nature.com/articles/s41467-019-08987-4.
> > And industrial data with known shortcuts such as the Covid-19 dataset of https://www.nature.com/articles/s42256-021-00338-7.pdf, the phenotyping dataset of https://www.nature.com/articles/s42256-020-0212-3 or the ISIC (skin cancer) dataset e.g. illustrated in Rieger et al. (ICML 2020, https://icml.cc/media/icml-2020/Slides/5914.pdf).
> >
> > I agree with Reviewer **C42n** that the current definition of shortcut learning in the present study is misleading "because it does not identify specific shortcuts and then demonstrate that it has reduced them."

---

> > > ### Author Response · Authors · 2021-11-21
> > > **Further Responses**
> > >
> > > Thank you for taking the time to give a further reply. Below please find our further responses to your concerns.
> > >
> > > **Unclear Motivation** We would like to clarify that we have given a pretty detailed motivation in this paper. Both the Reviewer w6iS
> > > >The paper is very well-written, especially the first three sections (introduction, motivation, etc.).
> > >
> > > and the Reviewer u8Nt
> > > >The current work is well motivated and very relevant
> > >
> > > have agreed that this paper is well-motivated.
> > >
> > > The description of motivation is mainly placed in the Introduction section(Page1-2). As we have described, we adopt the gradient starvation hypothesis[Gradient starvation: A learning proclivity in neural networks. NIPS2021] that shortcut learning can be attributed to the propensity for learning statistically dominant patterns that lack generalization information. The printer test page gives us the inspiration that we can draw lessons from the solution that were applied to the color cast problem. Therefore we choose to use the white paper as an indicator to detect dominant patterns. We hope this could clarify your concern and help you better understand the motivation.
> > >
> > > **Unclear contribution, why the authors do not show the performance of the proposed approach on already investigated shortcut problems**
> > >
> > > - You have already approved the progress we have made on the benchmark datasets, as in
> > > > it is a good first step to evaluate on benchmarks.
> > >
> > >   As you can see that the datasets in our paper are all common, general and frequently used datasets. Bearing the weight of a large number of researches, these datasets themselves hold considerable credibility. Our work, focusing on shortcuts, indeed achieves consistent performance boosts on these datasets. Therefore, we believe that a solid work that made such progress is worth sharing with the community.
> > >
> > > - There is an obvious contradiction in your logic that you think such a contribution is unclear and demand us to include results on more datasets. First, a fundamental consensus in shortcut learning is that the emergence of shortcuts is not limited to some so-called "already investigated shortcut" datasets. All the datasets may have unintended biases that a model training on these would rely on. With limited manpower, it is impossible to verify our effectiveness in all the datasets from all the fields. Using a synthetic conceptual experiment, WP has already demonstrated its ability on a concrete scenario that could be easily understood. Second, some of the datasets you listed, for example the Covid-19 dataset and the skin dataset, are specific. They are usually found in medical papers instead of papers in the field of general computer vision.
> > >
> > > Therefore, if you have found any technical flaws in our experimental design, we would accept them in good faith and correct them in the manuscript. But we believe this behavior right now is either unprofessional or you using your right of a reviewer indiscriminately.
> > >
> > > **It does not identify specific shortcuts and then demonstrate that it has reduced them**
> > > - First, concretizing the shortcut into a special scenario, our conceptual experiment is a clear proof that WP has the ability of alleviating the specific bias that we randomly designed.
> > > - Second, as we have explained in the responses to Reviewer C42n, what we describe shortcut in this paper did not restrict it to be specific. Most shortcuts are subtle and invisible to the human eye [1,2]. Besides, a specific shortcut represents specific knowledage, since it is based on a fundamental premise: a human must go through the datasets to statistically discover potential risk. On the contrary, WP has the ability of alleviating various shortcuts in various datasets, and more importantly, without any supervision/experience from human. Such the ability is more of an advantage than a weakness.
> > >
> > > [1] Shortcut Learning in Deep Neural Networks. Nature Machine Intelligence 2020
> > > [2] Invariant Risk Minimization. 2019

---

### Official Review · Reviewer_u8Nt · 2021-11-02

**Correctness:** 3
**Technical Novelty And Significance:** 4
**Empirical Novelty And Significance:** 3
**Recommendation:** 8
**Confidence:** 4

**Main Review:**

**Strength:**

The paper is easy to read and understand, and the arguments are mostly clear and concise. The prior works are explained well and the current work is well motivated and very relevant. The proposed method is simple and effective, the right combination for wide adoption, and the insights from Section 4 on how WP affects different parts of the CNN classifier can help improve the overall understanding of these models in the community. The experiments are well-designed and clear in what they try to illustrate, and sufficiently analyze the performance of the method, on its own and in conjunction with other related regularizations, as well as the contribution of different hyperparameters.

**Concerns and Questions:**

1- In Section 2, the paper argues that “instead of leveraging laborious and expensive supervision, our method utilizes the common sense by leveraging the white paper to detect the dominant patterns”, however, I cannot find any experiment/figure backing this up. Reporting some cost comparison (training time, computation, etc.) could help justify this claim.

2- Table 1 is very helpful in illustrating the effectiveness of WP, however when it comes to testing the method using other input types (e.g. noise or ice-cream), the paper only provides the overall performance on CIFAR100. A more clear and helpful comparison would be to also include these different input choices in Table 1.

3- Table 4 is confusing to me: why does the first row read “+White Paper Assistant” and then has “-” under “w/ WP”? Are these methods being stacked as we go down the rows of the table (for example is the last row using all the above methods together, or just Label smoothing w/ and w/o WP? I suspect it is the latter case, in which case I think “+” should be dropped from the rows to avoid confusion).

4- In a closely related line of work, adversarially-trained CNNs have been shown to be less reliant on shortcut features and focus on more “human-expected” features (see for example Figure 2 and 3 in [1]). In particular, these models can reduce error of classification on data limited regimes (such as the small datasets used in this paper). Including a study of whether WP achieves the same performance boost on adversarially-trained CNNs, and moreover, whether WP can be used as an alternative to adversarial training for robustness, would be very helpful for the community. But at the very least, I recommend referencing such works and commenting on potential effects.

5- An important consideration when deciding to use WP is the time/accuracy trade-off, that is, how will increasing the number of WP iterations (M) from zero to 50 or 500 affect training time, and in turn accuracy (this can be achieve by adding a training time plot to Figure 6 of Appendix B).

6- How will the size of the dataset affect the helpfulness of WP (as most of the datasets used in the paper seem to be on the smaller size)? This is an important point to consider since methods that try to avoid shortcut connections tend to degrade performance on larger datasets (see [1]).

Side note: Figure 3e should be 3d.

[1] Tsipras et al. Robustness may be at odds with accuracy. ICLR (2019).


**Summary Of The Paper:**

The paper proposes a novel method, White Paper Assistant (WP), to prevent CNNs from utilizing spurious input-out correlations, the so-called shortcuts, in classification. The main idea is to intermittently update the CNN to predict uniform distribution over classes for white image inputs. Through careful and extensive empirical studies on various datasets, the paper shows that this simple regularization can prevent the CNN from excessively focusing on shortcuts, thus learning more generalizable features and improving the overall accuracy on the test set.

**Summary Of The Review:**

I enjoyed reading this paper, and I think its proposed method is mostly effective and can be valuable as a new regularizer for CNNs. Moreover, the insights into how different parts of a CNN are affected by such a regularizer strikes me as a very helpful piece of knowledge for the wider research on neural net architectures. As such, despite having some concerns listed previously, I think the paper merits acceptance.

---

> ### Author Response · Authors · 2021-11-16
> **To Reviewer u8Nt**
>
> Sincerely thanks for your time and efforts. We are very grateful for your approval of our work. Below please find our responses to your concerns point by point.
>
> 1.**Sentences in Section 2** We would like to clarify that in the sentence "instead of leveraging laborious and expensive supervision, ......", we were giving a description of the differences between our method and other methods that remove shortcuts. For example, in the paper we listed before the sentence(Imagenet-trained CNNs are biased towards texture; increasing shape bias improves accuracy and robustness, ICLR2019), Geirhos et al. use style transfer technique to generate a new dataset (Stylized-ImageNet) where the texture information was randomly replaced. Also in the paper we listed(Shape-texture debiased neural network training, ICLR2021), lI et al. use a similar way to generate cue conflict images provide supervision from both shape and texture. Though being effective, these methods indeed leverage laborious(we need manually use style transfer to generate) and expensive(considerable computation cost) supervision. By comparison, our method utilizes white paper that only requires one line of code and no preliminary work.
>
> 2.**Revisions on Figure 2** Thanks again for your useful comments. We agree with you that when testing other input types, it would be more useful that we use the experimental settings as in Table 1 which is more directly linked with what WP is designed to do. We would revise that in our new manuscript. We provide partial experiment results here. When we use the Gaussian Paper Assistance on Shortcut-CIFAR100. The top-1 accuracy on Shortcut-CIFAR100 and CIFAR99 would be 66.17(66.73,65.94,66.20,65.82)and 43.92(45.26,41.71,43.17,45.53), respectively. These results are slightly weaker than the white paper but still competitive, highlighting the effectiveness of the detect-and-conquer algorithm.
>
> 3.**Revisions on Table 4** No, these methods were not stacked as we go down the rows. The results of each row are independent, representing the accuracy with or without the combination of WP.  The first row with“+White Paper Assistant”was to make it easier to compare the performance of each method(including WP) when used individually. Since it indeed leads to confusion, we will delete that in the revised manuscript.
>
> 4.**Adversarial Training** Thanks for your recommendation. The paper you listed is very illuminating. After a rough reading, there are several aspects worth describing. 1).The idea of WP and adversarial training is similar to some extent. The aim of adversarial training (forcing the classifier to rely on robust features) is similar to WP as well. So referencing such works is needed.  2)The adversarial training does improve accuracy on limited data regimes, but this positive effect has gradually disappeared on CIFAR-10(as in Figure 1 in the paper), where WP can still help. Therefore, it may be safe to say WP is less sensitive to the data volume compared with the adversarial training. But it is still interesting to see whether WP improves the performance on adversarially-trained CNNs. If positive, it may suggest that the adversarially-trained CNNs would still have bias/shortcut. We will work on that in the near future. 3)We find the co-occurrence of the improved accuracy and robustness in WP may support the saying "it is possible to obtain a robust classifier by directly training a standard model using only features that are relatively well-correlated with the label (without adversarial training)"(page 6). But the experimentation of this needs to be much more extensive in order to be more confident in this claim. A future version of this work would have more focus on this point.
>
> 5.**Time/accuracy discussion** We would add an accompanying training time plot below the Figure 4. Thanks for your suggestion. We have to admit that WP indeed has a time-accuracy trade-off. The best performance is achieved when the computation cost is close to that of real images training. But it is worth noting that WP can still achieve considerable performance boosts given a small budget of time.
>
> 6.**Larger datasets** Limited by the computational resources and time, we did not conduct experiments on larger datasets such as ImageNet-1k. But we believe experiments of the scale in the paper have given a good indication of the effectiveness of the method. Furthermore, considering that tiny-ImageNet is a miniature version of ImageNet-1k, we have strong confidence in the performance boost of our method in ImageNet-1k. We would try to include experiments on larger datasets in the future version of this work.

---

> > ### Comment · Reviewer_u8Nt · 2021-11-21
> > **Final Thoughts**
> >
> > Thank you for the explanations, I strongly recommend adding these to the main paper. In general, it is good practice to avoid umbrella statements like "laborious and expensive" and replace them with explicit statements like the ones you've provided in your response. I've read the other reviews as well, and while I agree with some of the concerns regarding the extent of experiments, I still think the contributions presented in this work are novel and valuable, therefore keeping my recommendation for acceptance.

---

### Official Review · Reviewer_w6iS · 2021-11-03

**Correctness:** 3
**Technical Novelty And Significance:** 1
**Empirical Novelty And Significance:** 3
**Recommendation:** 5
**Confidence:** 3

**Main Review:**

Strengths:
1. The paper is very well-written, especially the first three sections (introduction, motivation, etc.).
2. Authors conducted a very comprehensive set of empirical experiments and ablation studies to show the usefulness of WPA.
3. The idea is novel.

Weakness:
1. The idea is a bit simple -- which in of itself is not a true weakness. ResNet as an idea is not complicated at all. I find it disheartening that the paper did not really tell readers how to construct a white paper in section 3 (if I simply missed it, please let me know). However, the code in the supplementary materials helped. White paper is constructed as follow:
```python
white_paper_gen = torch.ones(args.train_batch, 3, 32, 32)
```
It offers another way of constructing white paper, which is
```python
white_paper_gen = 255 * np.ones((32, 32, 3), dtype=np.uint8)
white_paper_gen = Image.fromarray(white_paper_gen)
white_paper_gen = transforms.ToTensor()(white_paper_gen)
white_paper_gen = transforms.Normalize((0.4914, 0.4822, 0.4465), (0.2023, 0.1994, 0.2010))(white_paper_gen)
```
The code states that either version works similarly and does not affect the performance.
I wonder if there are other white papers as well, for example `np.zeros((32, 32, 3))` -- most CNN models add explicit bias terms in their CNN kernel. Would a different white paper reveal different bias in the model? I don't think the paper answers this question or discusses it.
2. Section 4 "Is white paper training harmful to the model?" -- the evidences do not seem to support the claim. The evidences are 1). Only projection head (CNN layers) are affected but not classification head (FCN layer); 2). Parameter changes are small. None of these constitute as a direct support that the training is not "harmful" to the model. This point can simply be illustrated by the experimental results
3. Section 5.1 and 5.2 mainly build the narrative that WPA improves the test performance (generalization performance), but they are indirect evidence to support that WPA does in fact alleviate shortcut learning. Only Section 5.3 and Table 6 directly show whether WPA does what it's designed to do. A suggestion is to discuss the result of Section 5.3 more.
4. It would be interesting to try to explain why WPA works -- with `np.ones` input, what is the model predicting? Would any input serve as white paper? Figure 2 seems to suggest that Gaussian noise input does not work as well as WPA. Why? Authors spend lot of time showing WPA improves the test performance of the original model, but fails to provide useful insights on how WPA works -- this is particularly important because it can spark future research directions.

**Summary Of The Paper:**

The paper proposes a method called White Paper Assistance that serves as a data/task agnostic regularization to prevent shortcut learning. It conducts a set of empirical experiments to show the usefulness of this technique.

**Summary Of The Review:**

The paper proposes a simple idea and conduct a wide range of experiments that seem to focus on producing SOTA results instead of illustrating how and why WPA works. However, given the amount of effort that goes into this paper, I'm on the fence of whether to recommend accept or reject.

---

> ### Author Response · Authors · 2021-11-15
> **To Reviewer w6iS**
>
> Sincerely thanks for your time and efforts. We are very grateful for your approval of our work. Below please find our responses to your concerns point by point.
> 1. **Two versions of white papers**. We really appreciate your approval for the simplicity of our work. We agree with you that detailed instructions for better reproducibility are needed. About your concerns, we first explain the reason for having such two versions. The direct and standard implementation of our method is the second version, wherein we first generate a white paper and then normalize it. But it is still not ready until we generate a batch of it. It is somewhat complex and time-consuming. Luckily, we find that the first version that only requires one line of code is able to achieve competitive performance. So, for the sake of simplicity and time, we use the first version. We will also add this description in our revised manuscript
> 2. **If there are other white papers**. If you mean the different ways to get a batch of white paper. Of course, for example, we can create a sheet of white image locally and then use the code to load this white paper. But, theoretically it would get the same result as the second version. PS, your example of ``np.zeros((32, 32, 3))`` would actually lead to a sheet of black paper. It is also working, and we will discuss it in the following responses(Responses 7).
> 3. **Would a different white paper reveal different bias in the model?** Yes. The core idea of our method is that we believe the model would have bias/preference. Thus we use the white paper to detect the bias. The bias that has been detected, actually denotes the unintended propensity that would make the model misidentify the white paper. So, when we use other papers, for example black paper, it would detect different bias, bias that makes the model misidentify the black paper .
> 4. **Subtitle of Section 4**. We agree with you that both evidences fail to serve as a direct support that the training is not "harmful" to the model. This part was originally acting as a demonstration of the characteristic of WP. We find it could link with the fact that WP is not harmful, since we were worried that one might consider WP being harmful due to the clear performance drop after WP is conducted. But it is indeed indirect, as you say. We would revise that in our manuscript.
> 5. **Discuss the result of Section 5.3 more**. We agree with you that Section 5.3 should be discussed more, since it is directly linked with the intention. However, we would like to note that we have provided a strong and direct experiment that could prove WP alleviates the shortcut learning in Section 3(Q1, page 3-4). The findings in the experiments have manifested the ability of WP to restrain the excessive reliance on dominant patterns.
> 6. **What is the model predicting.** As we claimed in the middle of page 3, when the white paper is fed into the model, the model would give us the output. The distribution of this output represents the perception of the model for this white paper, or to say, the extent to which certain pattern plays a dominant role for a class and lead the model to misclassify the white paper(page 5).
> 7. **Would any input serve as white paper?** Yes. One of the key ideas of the whole scheme is that the input images (whether they are white papers or colored papers) do not belong to any classes the model have learned from whichever benchmark dataset, hence a trained model should give an inference result that is almost as if it makes a random guess (page 3). Therefore, any images that could apply this logic would actually have an effect in detecting shortcuts. For example, when we use the black paper on ResNet 56, the performance is still quite good (74.33, 74.32, 74.51, 74.19, baseline73.66).
> 8.**Gaussian noise input does not work as well as WPA**. A possible explanation for this might be that the uninformative nature of the white paper seems to make it more suitable for detecting spurious dominant patterns, since the lack of semantics itself means no bias towards any pattern(page 5). In light of the perspective of information entropy, the gaussian noise is more formless/disordered compared with the white paper. Such property would probably lead to instability when detecting dominant patterns compared with the white paper. We would have more focus on this point in the future version of this work.
> 9. **Fail to provide useful insights on how WPA works**. We respectfully disagree with you that we did not spend efforts in discussing WP. We actually have a whole section named "HOW DOES WHITE PAPER ASSISTANCE WORK" in our paper to discuss the behavior of WP. We also designed an experiment to prove the effectiveness of WP in alleviating the shortcut learning and exhibited the possibility of other alternative paper in WP. All these parts(page3-6, page 8) link with WP directly. By comparison, the part(page6-7) where we show WP improves the performance did not take up too much weight.

---

### Decision · Program_Chairs · 2022-01-20

**Decision:**

Reject

**Comment:**

This paper studies the "shortcut" learning phenomenon in CNNs and proposes a simple and effective strategy (white paper) to alleviate specific shortcut patterns (e.g. "black squares" in the image). The proposed scheme is verified empirically and shown to improve over some existing solutions. All reviewers appreciate the simplicity of the idea, which allows its quick implementation and reproduciblity. However, reviewers y5Su and C42n believe the notion of shortcuts as studied in this paper are not only very limited, but also artificial. Consequently, they raise doubts about practical relevance/significance of the method for real world datasets with natural shortcuts. Based on these concerns, I suggest authors to identify a real setting (non-artificial data) where, alongside their synthetic shortcuts, can show the practical effectiveness of the proposed can.